# Retinal and Choroidal Thinning—A Predictor of Coronary Artery Occlusion?

**DOI:** 10.3390/diagnostics12082016

**Published:** 2022-08-20

**Authors:** Indrė Matulevičiūtė, Agnė Sidaraitė, Vacis Tatarūnas, Audronė Veikutienė, Olivija Dobilienė, Dalia Žaliūnienė

**Affiliations:** 1Department of Ophthalmology, Lithuanian University of Health Sciences, 44307 Kaunas, Lithuania; 2Institute of Cardiology, Lithuanian University of Health Sciences, 44307 Kaunas, Lithuania; 3Department of Cardiology, Lithuanian University of Health Sciences, 44307 Kaunas, Lithuania

**Keywords:** optical coherence tomography angiography, optical coherence tomography, foveal avascular zone, vascular density, coronary heart disease

## Abstract

*Introduction*. Optical coherence tomography (OCT) and optical coherence tomography angiography (OCTA) allowed visualization of retina and choroid to nearly the capillary level; however, the relationship between systemic macrovascular status and retinal microvascular changes is not yet known well. *Aim*. Our purpose was to assess the impact of retinal optical coherence tomography (OCT) and optical coherence tomography angiography (OCTA) parameters on prediction of coronary heart disease (CHD) in acute myocardial infarction (MI) and chronic three vessel disease (3VD) groups. *Methods*. This observational study included 184 patients—26 in 3VD, 76 in MI and 82 in healthy participants groups. Radial scans of the macula and OCTA scans of the central macula (superficial (SCP) and deep (DCP) capillary plexuses) were performed on all participants. All participants underwent coronary angiography. *Results*. Patients in MI groups showed decreased parafoveal total retinal thickness as well as GCL+ retinal thickness. Outer circle total retinal thickness and GCL+ retinal thickness were lowest in the 3VD group. The MI group had thinner, while 3VD the thinnest, choroid. A decrease in choroidal thickness and vascular density could predict 3VD. *Conclusions*. A decrease in retinal and choroidal thickness as well as decreased vascular density in the central retinal region may predict coronary artery disease. OCT and OCTA could be a significant, safe, and noninvasive tool for the prediction of coronary artery disease.

## 1. Introduction

Coronary heart disease (CHD) is the leading cause of death in developed countries. In addition, an expression of atherosclerosis in coronary arteries is the main cause of coronary heart disease [1]. Atherosclerosis is a chronic inflammatory condition, and it usually affects blood vessels all over the body. The retina is the only tissue of the human body that allows direct noninvasive visualization of microvasculature. The relations between various systemic diseases, including heart, and changes in ocular microvasculature has been a focus of studies in recent years [2,3,4]. Alternations in the thickness of the retina and its layers as well as changes in retinal microvasculature parameters was shown to be related to cardiovascular health [5,6,7]. Poplin et al., in 2018, developed an algorithm based on about 300,000 images of retinal fundus. The authors showed that the algorithm may determine cardiovascular risk factors with 70% of accuracy [8]. The obtained accuracy is similar to available cardiovascular disease risk calculators such as the SCORE (Systematic Coronary Risk Evaluation) [9]. However, the previously mentioned calculator requires additional tests and more profound evaluations of patient health factors.

The complexity of retinal structure and function has high oxygen demands and, thus, requires well-regulated blood flow [10]. Vascular disease cause mechanical trauma and degradation of certain glycocalyx components and by doing that impair endothelial barrier function and, eventually, cause capillary impairment. What is more the imbalance between superoxide and superoxide dismutase in retinal endothelial cells makes them more vulnerable to oxidative stress [11]. Reactive oxygen species (ROS) are integral part in the development of atherosclerosis and manifest CVD [12]. Therefore, it is not surprising that the retinal endothelium is affected by the presence of systemic cardiovascular (CV) risk [13]. The autoregulation of blood flow in the retina is maintained mainly by adaptations of retinal arterioles to changes in perfusion pressure (pressure autoregulation) [14] and metabolic needs (metabolic autoregulation) [13,15]. Metabolic factors are released by vascular endothelium and surrounding tissues. NO and prostacyclin (PGI2) are the factors that have relaxing effects, while endothelin-1 (ET-1), angiotensin II, and cyclooxygenase (COX) products, such as thromboxane-A2(TXA2) and prostaglandin H2 (PGH 2), have contracting effects. Extracellular lactate leads to the contraction or relaxation of the vessel wall, according to the metabolic needs of the tissue. The mechanical part of retinal vascular autoregulation involves glycocalix and ion channels in endothelial cells that transduces mechanical forces and enables endothelium to respond to them [11,16]. The vascular tone of retinal vessels is also sensitive to changes in O_2_ and CO_2_ and reacts to high levels of O_2_ by decreasing retinal blood supply [13,15,17].

Atherosclerosis, as a widespread condition [18], is usually diagnosed in the site of a specific lesion only, e.g., the heart in the case of coronary heart disease. There are several invasive and noninvasive techniques to diagnose atherosclerotic plaques. Invasive techniques such as intravascular ultrasound, optical coherence tomography, near infrared spectroscopy and their combined imaging modalities have certain advantages regarding the diagnosis of atherosclerotic plaques, but each has its own limitations. Therefore, multimodal imaging would be potentially more advantageous, even though it has its own weaknesses—its invasive nature, higher cost, inability to visualize entire coronary trees and inability to provide a complete assessment of coronary artery pathophysiology [19]. Noninvasive modalities (positron emission tomography (PET), computed tomographic coronary angiography (CTCA), and magnetic resonance imaging (MRI)), on the other hand, have their own specificities. PET is good at the identification and quantification of the inflammation of atherosclerotic plaque but requires a radiotracer and is expensive. CTCA has high predictive value, but requires radiation and a contrast agent to be injected. MRI provides detailed information on the artery wall and plaque composition but takes a long time to scan and is not suitable for patients with metal devices [19]. Nanoparticles for diagnostic and therapeutic purposes have been investigated recently as well. However, they do not work as a separate imaging tool and work as a probe for another previously mentioned imaging modality only.

Coronary angiography is an invasive method to diagnose and treat CHD via stent implantation, if necessary. Coronary angiography without angiographically visible obstruction is present in up to 20% of patients with ischemia due to microvascular disease [20,21,22,23]. Considering the disadvantages of all the aforementioned diagnostic modalities, a non-invasive and inexpensive diagnostic tool for atherosclerosis suspicion would be highly appreciated. Non-invasive visualization of retinal vessels may help identify significant microvascular dysfunction in myocardium and allow to predict risk of cardiovascular events [23,24,25]. Analysis of The Blue Mountains Eye Study and The Beaver Dam Eye Study showed that narrowing of arterioles and dilatation of venules is associated with a 40–70% higher risk of dying from MI in middle-aged patients [26]. The Eye-MI Study found a relationship between lower inner retinal density and peripheral arterial disease, CHD.

Developing techniques, such as swept source optical coherence tomography (OCT) and optical coherence tomography angiography (OCTA), have allowed segments of the retina and choroid to be visualized down to nearly the capillary level. Vessel diameter can be derived from structural OCT and OCTA data [27,28]. OCTA produces a depth-resolved, high-resolution display of retinal and choroidal vasculature without the use of intravenous contrast agents and dyes [29]. Microvascular OCTA imaging technology can provide an insight into the capillaries’ architecture. The structural changes in eye microvasculature may be the earliest markers of ischemia before vascular changes [30]. There are data about the effect of hypertension and diabetes on changes in microvasculature [30,31]. Arnould and co-authors found that vascular density measured on OCTA was associated with the cardiovascular risk profile and with impaired left ventricle ejection fraction in patients with a high-risk cardiovascular status [32]. Recent studies provide information about the relationship between subclinical atherosclerosis and certain ocular parameters (decreased choroidal thickness and foveal vascular density) derived from OCT and OCTA [33,34]. 

Thus, OCT and OCTA may be excellent prognostic markers to predict atherosclerosis and the risk of cardiovascular events without the further intervention normally required for coronary artery imaging. We hypothesized that the changes in retinal and choroidal parameters might differ in relation to the extent of atherosclerosis-affected coronary arteries.

The purpose of our study was to assess the impact of retinal OCT and OCTA parameters on CHD in acute myocardial infarction (MI) and chronic three vessel disease (3VD) groups. 

## 2. Materials and Methods

This observational study was performed in the Hospital of the Lithuanian University of Health Sciences Kaunas Clinics Department of Ophthalmology during the period between January 2019 and November 2021. All the participants had coronary angiography performed as an urgent procedure for MI or scheduled coronary angiography. All the participants were divided into three groups—MI, control group with unobstructed coronary arteries and 3VD with all three coronary arteries affected (without previous history of acute coronary syndrome or revascularization). Kaunas’ regional bioethics committee approved the research protocol. Written informed consent was obtained from participants to allow analysis of the collected data. 

Inclusion criteria for MI group was acute myocardial infarction caused by total coronary artery occlusion proved by coronary angiography (including STEMI and NSTEMI). Cardiologic investigation was performed by cardiologist in the Department of Cardiology. The ocular investigation was performed during the first five days after the event. Exclusion criteria for all groups were high refractive error (myopia and hyperopia greater than 6.0 diopters or astigmatism greater than +/− 3.0 diopters), amblyopia, previous ocular trauma, intraocular inflammation, ocular surgeries except cataract removal with uneventful phacoemulsification surgery, glaucoma, macular disorders or any conditions obscuring the view of the fundus. Patients with diabetes were not included into the study. The information about arterial hypertension and heart diseases, smoking, alcohol consumption, physical activity, body weight, and height was collected. Waist circumference was measured upon investigation, and echocardiography and blood tests were performed as a standard care procedure for cardiology patients. Alcohol consumption was converted into standard alcohol units by the investigator (1 unit–100 mL of wine (11–13%), 200 mL of beer (5%), 60 mL of wine (18%), 25 mL of strong spirit (40%)). During the investigation, a standard ophthalmological examination was performed: uncorrected and best corrected (according to the refraction) visual acuity using ETDRS chart from 4 m distance, Goldmann tonometry, biomicroscopy and ophthalmoscopy. The condition of the lens was graded according to the Lens Opacities Classification System III (LOCS III); if the cataract was removed, that was marked in the data. If any of the exclusion criteria were found during the examination, the patient was not included in the study.

OCT scans of the macula were taken using swept source OCT. DRI OCT Triton (Topcon, Tokyo, Japan) instrument uses a wavelength of 1050 nm and can acquire 100,000 A-scans per second. Radial scans of the central foveal region were performed, and central retinal and choroidal thickness were evaluated using automatic software procedure. Retinal thickness from internal limiting membrane to retinal pigment epithelium in the nine ETDRS regions (central, inner nasal, inner temporal, inner superior, inner inferior, outer nasal, outer temporal, outer superior and outer inferior) were calculated and were produced automatically. The average of inner (inner nasal, inner temporal, inner superior, inner inferior) and outer (outer nasal, outer temporal, outer superior and outer inferior) circle was calculated and represented in this article (Figure 1). 

The same procedure was performed to evaluate the choroidal thickness and retinal layers (RNFL (ILM-RNFL/GCL), GCL+ (RNFL/GCL-IPL/INL), and GCL++ (ILM-IPL/INL)). RNFL includes only the innermost retinal nerve fiber layer, GCL+ includes two separate retinal layers—ganglion cell layer and inner plexiform layer—and GCL++ measures all the aforementioned layers—GCL+ and RNFL—from the retinal nerve fiber layer to inner plexiform layer (Figure 2). 

OCT angiography was performed in macula and 3 × 3 and 6 × 6 mm images of central macular area were acquired. The OCTA scans of the superficial and deep capillary plexuses were generated separately using the automated software algorithm. Based on the preset parameters, the superficial network extends from 2.6 microns below the internal limiting membrane to 15.6 microns below the inner plexiform layer (IPL). The deep capillary network extends from 15.6 to 70.2 microns below the IPL. Density maps in the superficial and deep capillary plexuses were generated and values in the central, superior, inferior, nasal, and temporal regions were shown automatically. The average value of vascular density map surrounding center was calculated and used as a circle value in logistic regression analysis. Vascular density was expressed in percentages in the aforementioned regions. Using the acquired OCTA images, the parameters of the foveal avascular zone (FAZ) were measured manually in the same software. The area of the FAZ and maximum horizontal and vertical diameters of the FAZ were evaluated (Figure 3).

### Statistical Analysis

The data was not normally distributed; therefore, nonparametric Kruskal–Wallis and Mann–Whitney U tests were used for comparison of three and two groups, respectively (MI, healthy, 3VD groups). When multiple comparisons were performed, Bonferroni correction was used and only Bonferroni adjusted *p*-value is presented. Binary logistic regression was used to predict MI and 3VD. The values were expressed as median, range, and percentages. A *p*-value < 0.05 was considered statistically significant. To predict ischemic heart disease with margin error of 5% and confidence level of 95% according to the incidence of CHD worldwide (1.72% [36]), we calculated the sample size of 26 patients. We included that number of patients in 3VD group and increased the number of patients in other groups.

## 3. Results

In total, 184 patients were included in the study (enrollment and exclusion of the patients is presented in Figure 4). General and cardiovascular characteristics were similar in the presented three groups. Diastolic blood pressure and the number of smoking pack years (55.7% of participants were smoking) were significantly higher in the MI group than in other groups. Patients with MI had lower left ventricle ejection fraction than patients from other groups. All characteristics of the patients are presented in Table 1.

Total retinal thickness was significantly different between all three groups in outer circle and different between the healthy and MI groups in the inner circle. Evaluating the thickness of separate retinal layers, there was no significant differences between groups in RNFL and GCL++ layers. On the other hand, the thickness of GCL+ was statistically significantly different in all areas except the central. Choroidal thickness differed significantly mostly between the healthy and 3VD groups, as well as the MI and 3VD groups. A comparison of total retinal and choroidal thickness as well as layers of retina can be found in Appendix A.

When comparing the parameters of the foveal avascular zone (area, vertical and horizontal diameter in superficial and deep capillary plexuses) between groups, no significant differences (*p* > 0.05) were found (Appendix A).

When comparing vascular density in separate segments in superficial and deep capillary plexuses between the groups, there were almost no significant differences. Vascular density in superficial central (21.30 (10.87–28.09) vs. 18.43 (11.75–25.63), *p* = 0.027) and nasal (45.72 (40.41–51.03) vs. 45.21 (40.05–48.78), *p* = 0.034) areas were larger in healthy participants compared with the 3VD group in 6 × 6 images (Appendix A).

### Logistic Regression Model for Cardiovascular Disease

The univariate analysis for the prediction of MI and 3VD is presented in Table 2 (only significant data is presented). Most of the ophthalmic measurements, as presented in Table 3, show decreased odds of developing 3VD with an increasing value of the parameters (GCL+ inner and outer circle, choroidal thickness, vascular density in superficial capillary plexus (6 × 6), vascular density in deep capillary plexus (3 × 3) central area). RNFL layer inner circle, age and creatinine were significant for the development of 3VD. Total retinal thickness in the outer circle, GCL+ in the inner circle and parameters such as left ventricular dimensions at the end of diastole, body mass index, and waist circumference decreased the chance of MI development.

Outer circle choroidal thickness and central vascular density in SCP shows decreased odds, while creatinine concentration shows increased odd for development of 3VD (Table 3).

## 4. Discussion

In the study, we evaluated retinal and choroidal thickness, retinal vascular parameters (vascular density and foveal avascular zone dimensions) patients after cardiovascular investigation. There were three separate groups according to coronary artery angiography results and clinical state of the participant— the 3VD and MI groups with obstructed coronary arteries and healthy group. We found a significant decrease in retinal and choroidal thickness in both the 3VD and MI groups. Logistic regression analysis showed that ophthalmic parameters could be used to predict MI and 3VD development (Figure 5). To our knowledge, this is the first study where total ophthalmic macular investigation from retinal and choroidal thickness to OCT angiography and vascular density was described in control, MI and 3VD groups. The comparison of our results and the ones of other OCT and OCTA studies is presented in Table 4—none of these studies included choroidal parameters in their investigation.

After paired comparisons, we detected significant differences in retinal thickness in the outer ETDRS segments with the thickest retina in the healthy, and the thinnest in the 3VD, groups. According to the comparisons of different retinal layers, all the differences in retinal thickness come from GCL+ (the ganglion cell and inner plexiform layer) differences. Patients with affected coronary arteries (the MI or 3VD groups) have statistically significantly thinner GCL+ layers in comparison to healthy participants, except the central part. Logistic regression analysis in different segments—GCL+ inner and outer circle—showed decreased odds of developing 3VD and total retinal thickness in the outer circle and GCL+ in the inner circle showed decreased odds of developing MI with increasing values of thickness. Wang J. et al. found that retinal thicknesses in CHD patients in all eight regions were smaller than controls but results were not statistically significant [38]. Lee W.H. et al. found that a hypertensive group had a significantly thinner retina, including separate RNFL and GCL-IPL layers, when compared to controls. They also found a significantly increased foveal avascular zone and decreased vascular density in a hypertension group in comparison to healthy participants. The authors stated that a decrease in retinal thickness and changes in OCT angiography may be due to chronic effects of high blood pressure due to microcirculation, causing ischaemia and a decrease in retinal thickness [31]. Different studies by Cheng C. et al. and Ziegler T. et al. agreed, by proving that risk factors of coronary artery disease and atherosclerosis significantly affect not only major heart vessels but microcirculation, including retinal and choroidal vasculature, causing capillary rarefaction and limitation of blood flow through sparse capillary networks and, therefore, being responsible for vascular damage and a decrease in retinal thickness [39,40].

In our study, choroidal thickness is statistically significantly lowest in the 3VD group in almost all areas in comparison to MI and healthy participants, while there was no difference between healthy and MI patients. Univariate logistic regression analysis of choroidal thickness showed <1.0 OR to have 3VD. Arnould et al. measured subfoveal choroidal thickness in an elderly population with an average age of 81.9 years, investigated its relations with cardiovascular risk factors and, contrary to our results, did not find any relations. The evaluation of the cardiovascular state of the participants was performed using a self-filled questionnaire [41]. Kocamaz et al. found that subfoveal choroidal thickness was significantly thinner in the coronary artery disease group compared to the patients who are at risk of coronary artery disease (CAD) [42]. Aydin et al. compared choroidal thickness between healthy controls and four groups of patients with low, moderate, high, and very high (patients with CAD) risk of cardiovascular diseases. The study proved choroidal thickness to be significantly lower at the subfoveal location in all the study groups but in the nasal and at temporal quadrants of high and very high-risk (CAD) group [43]. Waghamare and others carried out a study where choroidal thickness was measured and compared in healthy controls and patients with one of the main CVD factors—hypertension. They found that choroidal thickness at all locations was significantly lower in the hypertensive group, and it had a significant negative correlation with the systolic blood pressure and the duration of hypertension [44]. The average choroidal thickness of our participants is like Chinese healthy contemporaries’ in a study carried out by Ding and others. Moreover, choroidal thickness was found to decrease by 5.40 μm each year in 60-year and older populations [45]. This corresponds to Arnould L. et al.’s study results, where older participants were investigated and the thickness of the subfoveal choroid was decreased compared to our study [41]. Ahmad M. et al. performed a study where choroidal thickness was evaluated in coronary artery disease—defined as a history of >50% obstruction in at least one coronary artery on cardiac catheterization, a positive stress test, ST elevation myocardial infarction, or revascularization procedure—patients. They found significantly lower choroidal thickness in participants with coronary artery disease in subfoveal and surrounding choroid [28]. All of the above-mentioned studies correspond to our results, confirming that CAD has a negative effect on choroidal thickness.

We found that central and nasal vascular density in the superficial capillary plexus was significantly thinner in the 3VD group in comparison to healthy participants and results were confirmed by univariate and multivariate logistic regression analysis (OR for 3VD formation 0.735–0.872, *p* < 0.005). Aschauer J. examined coronary heart disease patients with a control group and findings were similar. They observed a trend of reduced VD in the total retina, as well as the individual retinal capillary plexuses (superficial and deep) and the choriocapillaris layer in patients with significant CHD, but neither these results nor logistic regression analysis were significant [37]. Wang et al. also reported that CAD patients have significantly decreased VD [38]. Zhong and others, in their study, determined the association between retinal microvasculature and the presence and severity of coronary artery disease (CAD): participants showed significantly greater odds of having CAD in the lower versus higher vascular density for mean superficial and deep capillary plexus [7]. Zhong et al. even developed a nomogram which helps to evaluate potential CAD patients before coronary angiography. It included the vessel density of the nasal and temporal perifovea in the superficial capillary plexus and vessel density of the inferior parafovea in the deep capillary plexus [46].

Even though we did not find any significant differences in FAZ parameters, a tendency of increased FAZ size is noted, especially in the 3VD group, which was also described in a study by Aschauer J. et al. [46]. In the review performed by Monteiro-Henriques and others, the authors concluded that patients affected by various cardiovascular diseases have generally lower vessel density and increased FAZ area [47]. These conclusions correspond to our results. The reason for insignificant results might be the lower number of participants in the 3VD group. The control group included in the study was engaged from the cardiology department, so they have cardiologic pathology, e.g., controlled hypertension or hypercholesterolemia, even though coronary angiography was clear.

It is important to signify hypercholesterolemia as a predictive factor for the development of myocardial infarction. Univariate logistic regression analysis showed that an increased unit of blood cholesterol increases odds of MI development by 1.4, and LDL cholesterol by 1.5. Cholesterol is a well-known factor for atherosclerotic plaque development in arteries [48]. Retinal cholesterol homeostasis dysfunction is important in deposit formation around retinal pigment epithelium and age-related macular degeneration development [49]. According to Tserentsoodol et al., cholesterol in the retina is not only locally synthesized but also up taken from systemic circulation, mostly via LDL cholesterol [50]. ApoB lipoprotein, as a major lipoprotein in LDL cholesterol, was investigated in retinal function. Shi et al. investigated type-2 diabetic patients without retinopathy and found decreased RNFL and GCL layer thickness as well as vascular density in hypercholesterolemic patients [51]. Age-related macular degeneration has been linked to CVD as well–drusen and atherosclerotic plaques have been proved to have several components in common, lipoproteins included [52].

All the aforementioned results enable the same conclusion—coronary heart disease is a sign of systemic atherosclerosis. Vascular endothelium damage is the sight of the initial atherosclerosis development [53]. Affected endothelium is unable to maintain vascular diameter, tone and blood flow (due to reduced nitric oxide and increased reactive oxygen species, as well as endothelin receptor-1) [54]. On the other hand, hyperlipidemia acts as one of the main risk factors of coronary heart disease. It causes lipid deposition in the vessels and a reduction in angiogenic sprouting. The process occurs via a downregulation of vascular endothelial growth factor (VEGF-A) and an additional reduction in endothelial N-Cadherin, the key anchoring protein with which endothelial cells and pericytes interact [55,56]. These changes cause previously described retinal and choroidal changes in the central retina—a reduction in retinal thickness (especially GCL, IPL, and INL, where superficial and deep retinal vascular plexus are located) [57] as well as choroidal thickness due to decreased blood flow, an increase in FAZ size and a decrease in vascular density because of downregulation of VEGF-A.

It is important to notice that, even though sex distribution did not show a statistically significant difference, there is a tendency towards a larger part of males in the groups of MI and 3VD in comparison to healthy ones, causing some inaccuracies. We also found that body mass index and waist circumference decreased the chance of MI development and this result is opposite to the clinical studies related to metabolic syndrome and cardiovascular disease [58,59]. This result might be due to uneven groups regarding BMI and waist circumference, where MI groups showed lowest results in comparison to other groups.

## 5. Conclusions

To conclude, our study showed that retinal thickness in the outer circle significantly decreases when coronary arteries are affected—the bigger the lesion, the thinner the retina. The most affected retinal layers are the ganglion cell layer and inner plexiform layer, forming GCL+. Significantly decreased choroidal thickness in all areas (central, inner, and outer circle) might be a sign of extensive coronary artery disease (3VD in our case). The results of OCT angiography showed that lowering of vascular density is a sign of increased odds of 3VD formation.

Considering all the results, we could conclude that 3VD patients with all coronary vessels extensively affected by atherosclerosis shows significant changes in retinal and choroidal microstructure as well. OCT and OCTA could be a significant, safe, and noninvasive tool in coronary artery disease diagnostics.

## Figures and Tables

**Figure 1 diagnostics-12-02016-f001:**
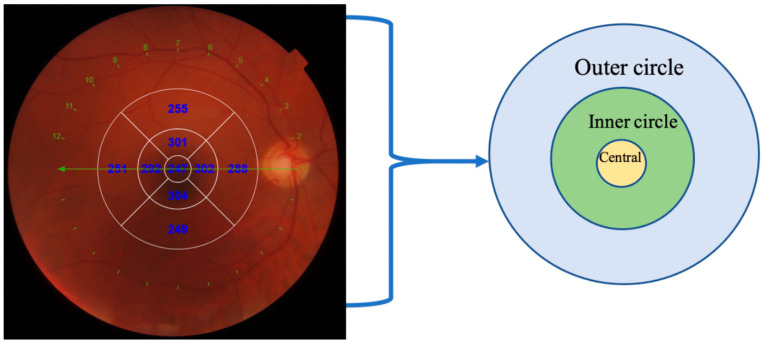
Calculation of retinal and choroidal thickness.

**Figure 2 diagnostics-12-02016-f002:**
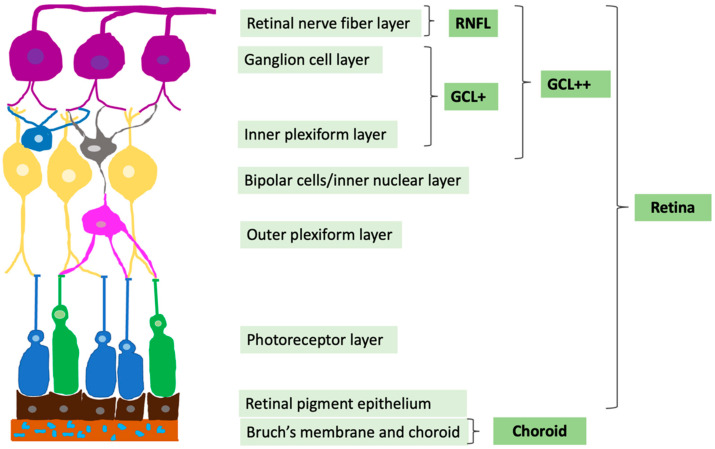
Representation of retinal layers evaluated by DRI OCT Triton. Adaptation from Vilades E. et al. 2020 [35].

**Figure 3 diagnostics-12-02016-f003:**
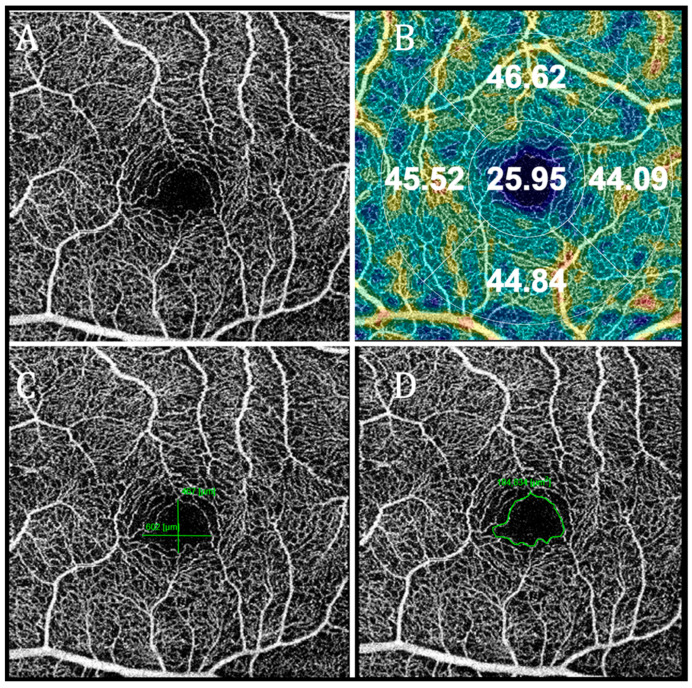
Images of optical coherence tomography angiography of superficial capillary plexus. (**A**)—automatically formed image; (**B**)—automatically formed image of vascular density values; (**C**)—manually drawn vertical and horizontal length of foveal avascular zone; and (**D**)—manually outlined foveal avascular zone.

**Figure 4 diagnostics-12-02016-f004:**
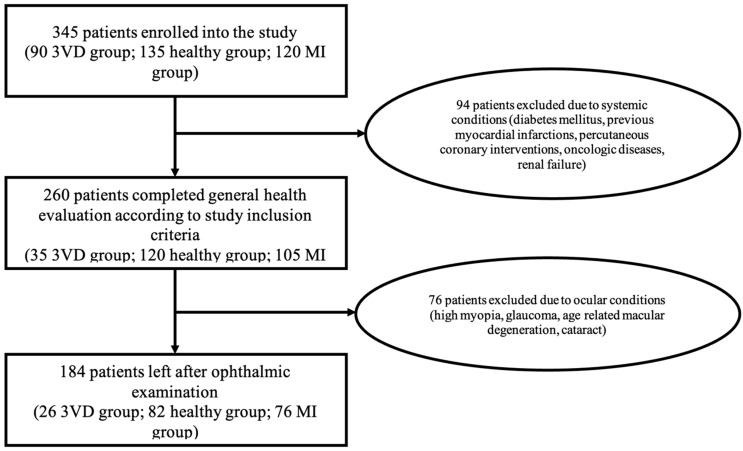
Patient enrollment. (MI—myocardial infarction, 3VD—three vessel disease group).

**Figure 5 diagnostics-12-02016-f005:**
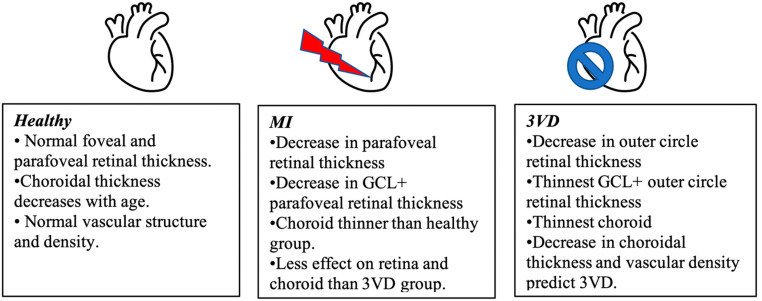
Main outcomes of the study.

**Table 1 diagnostics-12-02016-t001:** General and cardiovascular characteristics of all participants.

Variable	Healthy	Myocardial Infarction Group	3 Vessel Disease Group
Age in years			
Median (range)	61.22 (44.51–77.19) ^1^	61.76 (37.71–78.81)	64.83 (50.14–80.45) ^1^
Sex:			
Male, 112 (60.9%)	46 (56%)	48 (63%)	18 (69%)
Female, 72 (39.1%)	36 (44%)	28 (37%)	8 (31%)
Systolic blood pressure (mmHg)			
Median (range)	130 (100–180)	140 (80–250)	137.50 (120–160)
Diastolic blood pressure (mmHg)			
Median (range) *	80 (60–102) ^2^	85 (60–140) ^2^	80 (70–100)
Body mass index (kg/m^2^)			
Median (range) º	29.74 (23.03–45.91) ^3^	28.41 (20.55–39.79) ^3^	28.82 (22.07–36.78)
Waist circumference (cm)			
Median (range)	104 (81–132)	98 (70–133)	101 (76–131)
Smoking (in pack years)			
Median (range) •	12.13 (0.1–52.0) ^4^	26.75 (0.25–110) ^4^	23.5 (1.0–50.0)
Alcohol consumption (standard alcohol unit)			
Median (range)	0.75 (0.0–20.0)	0.5 (0.0–32.0)	0.0 (0.0–21.0)
Left ventricular dimensions at end of diastole (mm)			
Median (range)	49.0 (40.0–73.0)	48.0 (37.0–61.0)	48.5 (37.0–59.0)
Left ventricular posterior wall thickness at end of diastole (mm)			
Median (range)	11.0 (8.0–14.0)	10.0 (8.0–13.8)	10.0 (8.0–13.0)
Left ventricular ejection fraction (%)			
Median (range) •	55.0 (25.0–70.0) ^5^	45.0 (20.0–55.0) ^5,6^	55.0 (40.0–60.0) ^6^
Interventricular septum at end diastole (mm)			
Median (range)	11.0 (8.0–15.5)	11.3 (7.5–17.3)	11.0 (9.5–16.0)
Myocardial mass index (g/m^2^)			
Median (range)	97.80 (70.12–182.7)	95.42 (47.91–176.06)	91.56 (65.05–150.83)

* significant difference between healthy, MI and 3VD groups, *p* = 0.032; º significant difference between healthy, MI and 3VD groups, *p* = 0.046; • significant difference between healthy, MI and 3VD groups, *p* < 0.001; *1*—*p* = 0.029*; 2*—*p* = 0.038 (adjusted); *3*—*p* = 0.041 (adjusted); *4*—*p* < 0.001 (adjusted); 5—*p* < 0.001 (adjusted); *6*—*p* < 0.001 (adjusted), the number next to the value represents groups being compared.

**Table 2 diagnostics-12-02016-t002:** Univariate binary regression analysis for development of three vessel disease and acute myocardial infarction.

Variable	Myocardial Infarction			Three Vessel Disease		
	Odds Ratio	95% CI	*p*-Value	Odds Ratio	95% CI	*p*-Value
Body mass index	0.899	0.832–0.971	0.007	No significance		
Waist circumference	0.973	0.948–0.999	0.041	No significance		
Left ventricular dimensions at end of diastole (mm)	0.934	0.875–0.996	0.038	No significance		
Total cholesterol	1.366	1.026 1.818	0.033	No significance		
LDL cholesterol	1.503	1.051–2.149	0.026	No significance		
Atherogenic coefficient	1.523	1.128–2.055	0.006	No significance		
Age	No significance			1.064	1.008–1.123	0.024
Creatinine	No significance			1.034	1.00–1.070	0.047
**Retinal thickness**						
Outer circle	0.975	0.953–0.998	0.036	No significance		
**RNFL layer**						
Inner circle	No significance			1.082	1.004–1.166	0.039
**GCL+ layer**						
Inner circle	0.952	0.916–0.989	0.012	0.929	0.878–0.983	0.011
Outer circle	No significance			0.865	0.794–0.941	0.001
**Choroidal thickness**						
Central	No significance			0.993	0.986–1.00	0.041
Inner circle	No significance			0.989	0.982–0.997	0.005
Outer circle	No significance			0.987	0.978–0.995	0.002
**Vascular density—Superficial capillary plexus (FAZ 6 × 6)**						
Central	No significance			0.872	0.773–0.985	0.027
Circle	No significance			0.735	0.548–0.984	0.038
**Vascular density—Deep capillary plexus (FAZ 3 × 3)**						
Circle	No significance			0.773	0.602–0.992	0.043

**Table 3 diagnostics-12-02016-t003:** Multivariate logistic regression to predict three vessel disease.

Variable	Odds Ratio	95% CI	*p*-Value
Choroid outer circle	0.979	0.966–0.992	0.002
Vascular density—superficial capillary plexus (FAZ 6 × 6) central	0.819	0.699–0.959	0.013
Creatinine	1.041	1.00–1.084	0.048

**Table 4 diagnostics-12-02016-t004:** OCT and OCTA results of the studies investigating coronary heart disease.

Authors	Journal	Year of Publication	No. of Participants	Results
	Presented study			Choroidal thickness and central vascular density in SCP are significant predictors of three vessel disease. Decreased outer retinal thickness in MI and 3VD groups.
Zhong et al. [7]	Acta ophthalmologica	2022	410 participants	Decreased VD in SCP and DCP decreased total retinal thickness in coronary artery disease.
Aschauer et al. [37]	Transaltional Vision Science and Technology	2021	45 participants	A trend of decreased vascular density (*p* > 0.05) in coronary heart disease.
Wang et al. [38]	Biomedical optics express	2019	316 participants	Decreased retinal thickness, density and flow area, except fovea, and more intensive vessel density in outer retina in coronary heart disease patients. Retinal and choroidal microvasculature changes related to coronary artery and branch stenosis.
Arnould et al. [22]	Investigative ophthalmology and visual science	2018	275 participants	Decreased retinal vascular density (inner vessel density), association between inner vessel density and the GRACE and REACH score.

## Data Availability

Not applicable.

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
