# Peer review of "Retinal and Choroidal Thinning—A Predictor of Coronary Artery Occlusion?"

_diagnostics, 2022, doi:10.3390/diagnostics12082016_

Round 1

Reviewer 1 Report

This study evaluated the predictive power of OCT angiography in relation to coronary heart disease. The results support the potential value of this non-invasive technique for predicting heart disease.

A major weakness of the study is the limited number of patients, especially in the category of 3V. Actually, there is no mention of how the power of the study was determined apriori, and consequently, the sample size was not properly estimated. Another statistical weakness lies in the fact that no correction of p values was reported for multiple comparisons, resulting in an overestimation of the significance of the results.

The authors are encouraged to increase the patient sample size and to improve the statistical analysis.

Author Response

  • A major weakness of the study is the limited number of patients, especially in the category of 3V. Actually, there is no mention of how the power of the study was determined apriori, and consequently, the sample size was not properly estimated.

Thank you for the comment. To predict ischaemic heart disease with margin error of 5 % and confidence level of 95 % according to the incidence of CHD worldwide (1.72 %) [Khan MA, et al. 2020] we calculated the sample size of 26 patients. We included that number of patients in 3VD group and increased the number of patients in other groups.

  • Another statistical weakness lies in the fact that no correction of p values was reported for multiple comparisons, resulting in an overestimation of the significance of the results. The authors are encouraged to increase the patient sample size and to improve the statistical analysis.

Thank you for the comment and suggestions. In the calculations, where multiple comparisons were performed, the previously presented p value was changed to Bonferroni adjusted p value (table 1). The statement regarding adjustment was included in the statistical analysis section (lines 193-194).

As previously mentioned, we have included sample size determination statement in the paper (lines 196-200).  The patient enrollment scheme is presented in figure 4. The primary number of enrolled 3VD patients was in accordance to other groups, however to prevent the impact of other retinal and systemic diseases, we have excluded the patients with ocular conditions affecting foveal region and systemic diseases such as diabetes mellitus, renal, oncologic diseases and previous coronary interventions. As we have mentioned in the manuscript, CVD and age related macular degeneration (AMD) have some components of pathogenesis in common (lines 338-347). What is more cardiovascular diseases are related to metabolic syndrome and diabetes mellitus (Huang PL, 2009) as well and they are more pronounced in older population (Kuk JL, Ardern CI, 2010), so there were quite a lot of patients that were excluded due to previously mentioned conditions. AMD and CVD have both increased prevalence with the age and the higher extent of cardiovascular system envolvement in 3VD group makes it more difficult to find patients without retinal damage in that group of patients.

Reviewer 2 Report

In this paper, Indrė Matulevičiūtė et al evaluated a relationship between  OCT and OCTA parameters as predictors of coronary heart disease. The idea is innovative and elegant, but sometimes it is very hard to read the length of the sentences. I believe it can be improved. Here are my comments:

It is unusual to find an abstract without a division in introduction, methods, results and discussion. Please do this for clarity

There are many points in the citations that need citations (ex. lines 49,70,90)

I suggest shorting the sentences in the text because they are too long, and you lose the sense of the concept.

The authors mention very little about cholesterol, except in the table; However, this is the major risk factor for atherosclerosis; having excluded patients with diabetes mellitus and renal failure from the study, I believe that an extensive comment on cholesterol in the discussion is mandatory.

When the authors talk about MI they assert that a closed coronary artery has been found, so are they all STEMI infarcts, or have you also collected NSTEMI?

Please fix table 2 that is not clear (Perhaps a pagination error)

Author Response

  • It is unusual to find an abstract without a division in introduction, methods, results and discussion. Please do this for clarity

Thank you for pointing this out. We have added the section names to the abstract.

  • There are many points in the citations that need citations (ex. lines 49,70,90)

Thank you for the comment. We have added additional citations to the aforementioned lines:

49 line. Weiter JJ, Zuckerman R. The influence of the photoreceptor-RPE complex on the inner retina. An explanation for the beneficial effects of photocoagulation. Ophthalmology. 1980 Nov;87(11):1133-9.

70 line. Keeter WC, Ma S, Stahr N, Moriarty AK, Galkina EV. Atherosclerosis and multi-organ-associated pathologies. Semin Immunopathol. 2022 May;44(3):363-374.

90 line. Farrehi PM, Bernstein SJ, Rasak M, Dabbous SA, Stomel RJ, Eagle KA, Rubenfire M. Frequency of negative coronary arteriographic findings in patients with chest pain is related to community practice patterns. Am J Manag Care. 2002 Jul;8(7):643-8.

  • I suggest shorting the sentences in the text because they are too long, and you lose the sense of the concept.

Thank you for this suggestion. We tried to make the manuscript as clear as possible and there are quite a few places all across the paper were the formulations of the sentences have been changed to simplify the reading process and make the concept easier to understand. Everything is marked up in “Track changes”.

  • The authors mention very little about cholesterol, except in the table; However, this is the major risk factor for atherosclerosis; having excluded patients with diabetes mellitus and renal failure from the study, I believe that an extensive comment on cholesterol in the discussion is mandatory.

Thank you for the comment. We have added a section in the Discussion, regarding relations between cardiovascular disease, cholesterol, and eye. Lines 332-345.

  • When the authors talk about MI they assert that a closed coronary artery has been found, so are they all STEMI infarcts, or have you also collected NSTEMI?

Thank you for the question. We have added STEMI (70 patients) and NSTEMI (16 patients) patients in the study. 15 out of 16 NSTEMI patients had at least one segment of coronary arteries occluded by 90 % or more and only one had obstruction of 75 %. A comment in line 131 was added to the manuscript as well.

  • Please fix table 2 that is not clear (Perhaps a pagination error)

Thank you for the comment. We have rechecked table 2 and added additional comments “No significance” where no statistically significant differences were observed.

Round 2

Reviewer 1 Report

My comments have been fully addressed. No further comments.

Reviewer 2 Report

My answers are been addressed